# Rate-Optimal Online Convex Optimization
# in Adaptive Linear Control

Asaf Cassel[*]        Alon Cohen[†]        Tomer Koren[‡]

## Abstract

We consider the problem of controlling an unknown linear dynamical system under adversarially changing convex costs and full feedback of both the state and cost function. We present the first computationally-efficient algorithm that attains an optimal $\sqrt{T}$-regret rate compared to the best stabilizing linear controller in hindsight, while avoiding stringent assumptions on the costs such as strong convexity. Our approach is based on a careful design of non-convex lower confidence bounds for the online costs, and uses a novel technique for computationally-efficient regret minimization of these bounds that leverages their particular non-convex structure.

## 1  Introduction

We study a general setting of online adaptive linear control, where a learner attempts to stabilize an initially unknown discrete-time linear dynamical system while minimizing its cumulative cost with respect to an arbitrary sequence of convex loss functions. The system dynamics evolve according to

$$x_{t+1} = A_\star x_t + B_\star u_t + w_t,$$

where $x_t \in \mathbb{R}^{d_x}$, $u_t \in \mathbb{R}^{d_u}$ are the (fully observable) system's state and learner's control at time step $t$, and $w_t \in \mathbb{R}^{d_x}$ is the system noise added at step $t$ which is a zero-mean i.i.d. Gaussian random variable. The matrices $A_\star \in \mathbb{R}^{d_x \times d_x}$ and $B_\star \in \mathbb{R}^{d_x \times d_u}$ are the system parameters, which are assumed to be unknown ahead of time and need to be learned adaptively. The goal is to minimize regret with respect to a sequence of convex loss functions $c_1, \ldots, c_T$ over $T$ time steps, namely, the difference between the learner's cumulative control cost $\sum_{t=1}^T c_t(x_t, u_t)$ and the best cumulative cost achieved by a control policy from a given set of benchmark policies.

This general framework encapsulates numerous variations of learning in linear control that have been studied extensively in the literature. When the system parameters are known ahead of time and the costs are fixed and known (convex) quadratics, this amounts to the classical "planning" formulation of linear-quadratic (LQ) stochastic control; see [12]. The special case where the costs are fixed and known quadratics but the system parameters are unknown has been addressed much more recently [1, 21, 31]. This was recently extended to allow for a fixed and known convex cost [33] and later for stochastic i.i.d. costs [15]. On the other hand, the case where the system parameters are known but the quadratic costs are allowed to vary arbitrarily between rounds was first addressed in [20], and has been later extended in various ways to allow for arbitrarily-varying convex costs [5, 6, 36, 13]. In all of these special cases, we now know of efficient algorithms with rate-optimal $\sqrt{T}$ regret guarantees.

For the online adaptive linear control problem in its full generality, however, no regret-optimal algorithms are presently known. The state-of-the-art is due to [36] that achieved $T^{2/3}$-regret using a

---

[*]Blavatnik School of Computer Science, Tel Aviv University; `acassel@mail.tau.ac.il`.

[†]School of Electrical Engineering, Tel Aviv University, and Google Tel Aviv; `alonco@tauex.tau.ac.il`.

[‡]Blavatnik School of Computer Science, Tel Aviv University, Google Tel Aviv; `tkoren@tauex.tau.ac.il`.

36th Conference on Neural Information Processing Systems (NeurIPS 2022).

simple explore-then-exploit strategy: in the exploration phase, their algorithm estimates the dynamics parameters by exciting the system with noise; then, in the exploitation phase it runs an online procedure for known dynamics using the estimated transitions. This simple strategy has also been shown to achieve the optimal $\sqrt{T}$-regret when the online costs are additionally *strongly convex*, demonstrating that the stringent strong convexity assumption allows one to circumvent the challenge of balancing exploration and exploitation in online adaptive linear control.

In this paper, we resolve this gap and give the first rate-optimal algorithm for the general online adaptive linear control problem, accommodating arbitrarily changing general convex (and Lipschitz) costs and unknown system parameters. Our algorithm is computationally efficient and attains a $\sqrt{T}$ regret guarantee with polynomial dependence on the natural parameters of the problem.

**Techniques.** Our approach builds upon a combination of recent techniques in online linear control. First, we rely on the Disturbance Action Policies (DAPs) of Agarwal et al. [5]: our algorithm generates DAPs that choose the control at each time step as a linear transformation of past noise terms; the DAPs themselves are maintained by online convex optimization algorithms that generate slowly-changing decisions for guaranteeing the stability of the system throughout the learning process. Moreover, since the dynamics are unknown, our algorithm estimates the noise terms on-the-fly, and uses these estimates in place of the true noise vectors (this is akin to a technique in [33]).

Second, following the recent developments of Cassel et al. [15] for the case of stochastic costs, we perform regret minimization with respect to optimistic lower confidence bounds of the online costs. However, these confidence bounds turn out to be inherently nonconvex. To maintain computational efficiency, we adapt a trick of Dani et al. [22] (in the context of stochastic linear bandits) for relaxing the nonconvex objectives so as to assume the form of a minimum of a small number of convex objectives; then, we hedge over multiple copies of online gradient descent as "experts" in a meta-algorithm, where each copy minimizes regret with respect to one of these convex objectives.

Even so, the decisions of the hedging meta-algorithm are random and can thus change abruptly, interfering with the slowly-moving nature of the DAPs that is crucial for the stability of the system. We address this issue by using a lazy version of Follow the Perturbed Leader in place of the meta-algorithm (due to [7]) that employs only a small number of switches between experts. Overall, this results in a computationally efficient scheme that maintains the $\sqrt{T}$ regret rate of the individual gradient-based experts.

**Related work.** The problem of adaptive linear-quadratic control has a long history [e.g., 12]. Recent years have seen a renewed interest in this problem through the modern view of regret minimization—building on classic asymptotic results to obtain finite-time guarantees [1, 3, 9, 23, 24, 27, 32]. More recently [21, 31] provided polynomial-time algorithms obtaining an optimal $\sqrt{T}$ regret rate. The optimality of the $\sqrt{T}$ rate was proved concurrently by [14, 35].

More recently, [33] gave an efficient algorithm with $\sqrt{T}$ regret for learning the dynamics under a fixed known convex cost. [33] also observed that the problem of learning both dynamics and stochastic convex costs under bandit feedback is reducible to an instance of stochastic bandit convex optimization for which complex, yet polynomial-time, generic algorithms exist [4]. Cassel et al. [15] later study the problem of learning the dynamics and stochastic convex costs under full-information feedback. Unlike the approach of [33], their algorithm is based on an "optimism in the face of uncertainty" principle and is thus conceptually simpler and more efficient to implement.

Our approach relies on the standard assumption that the controller is provided with some initial stabilizing policy. First proposed in [23], such an assumption yields regret that is polynomial in the problem dimensions, and was later shown to be necessary by [18].

Past work has also considered adaptive LQG control, namely linear-quadratic control under partial observability of the state [for example, 36]. However, it turned out that in the stochastic setting, learning the optimal partial-observation linear controller is in a sense easier than learning the full-observation controller. It is in fact possible to obtain $\text{poly}(\log T)$ regret for adaptive LQG [29]. This result is facilitated by simplifying assumptions on both the noise distribution as well as the benchmark policy, assumptions which we do not make in this work.

Most works on regret minimization in adaptive control are model-based; meaning, the algorithm attempts to estimate the model parameters. Previous literature also considered the alternative approach

of model-free control [e.g., 2, 17, 25, 30, 37]. These works, however, rely heavily on the assumption of quadratic strongly-convex costs and do not apply to general convex costs.

Lastly, [13, 26, 33] consider control under bandit feedback. These results are unfortunately impeded by the state-of-the-art in Bandit Convex Optimization, that is either not efficient in practice (namely, high-degree polynomial runtime) or requires further assumptions on the curvature of the cost functions. For this reason we focus here on full-information feedback, with the hope that our techniques can be adapted to bandit feedback in subsequent work, contingent on future advancements in BCO.

## 2 Preliminaries

### 2.1 Linear control background

A discrete-time linear control system is one whose dynamics are governed by the following rule:

$$x_{t+1} = A_\star x_t + B_\star u_t + w_t,$$

where $A_\star \in \mathbb{R}^{d_x \times d_x}$, $B_\star \in \mathbb{R}^{d_x \times d_u}$, and where $w_t \in \mathbb{R}^{d_x}$ is zero-mean i.i.d. In the planning version of the problem the controller knows $A_\star, B_\star$ and, at each time $t$, can choose $u_t$ as a function of $x_1, \ldots, x_t$. After choosing $u_t$, the controller incurs a known cost $c(x_t, u_t)$. Classic results pertain to quadratic costs, and state that the control rule that minimizes the steady state cost $J(\pi) = \lim_{T \to \infty} \mathbb{E}_\pi [\frac{1}{T} \sum_{t=1}^T c(x_t, u_t)]$, chooses $u_t = Kx_t$ for some matrix $K \in \mathbb{R}^{d_u \times d_x}$. Moreover, the optimal rule $\pi_\star$ stabilizes the system, implying that $J(\pi_\star)$ is finite and well-defined for any quadratic cost function.

We require the following notion of strong stability [20], which is standard in the literature and whose purpose is to quantify the classic notion of (asymptotic) stability.

**Definition 1 (Strong stability).** A controller $K$ for the system $(A_\star, B_\star)$ is $(\kappa, \gamma)$−strongly stable $(\kappa \geq 1, 0 < \gamma \leq 1)$ if there exist matrices $Q, L$ such that $A_\star + B_\star K = QLQ^{-1}$, $\|L\| \leq 1 - \gamma$, and $\|K\|, \|Q\|\|Q^{-1}\| \leq \kappa$.

### 2.2 Problem setup

We address the problem of controlling an unknown linear dynamical system subject to general adversarial convex costs with full state and cost observation. In particular, the system parameters $A_\star$, $B_\star$ are initially unknown and the learner repeatedly interacts with the system as follows:

(1) The player observes state $x_t$;
(2) The player chooses control $u_t$;
(3) The player observes the cost function $c_t : \mathbb{R}^{d_x} \times \mathbb{R}^{d_u} \to \mathbb{R}$, and incurs cost $c_t(x_t, u_t)$.

Note that $(w_t)_{t=1}^\infty$ are unobserved, and the cost $c_t$ is revealed only after selecting $u_t$. Our goal is to minimize regret with respect to any policy $\pi$ in a benchmark policy class $\Pi$. To that end, denote by $x_t^\pi, u_t^\pi$ the state and action sequence resulting when following a policy $\pi$; then the regret compared to $\pi$ is defined as

$$\text{regret}_T(\pi) = \sum_{t=1}^T c_t(x_t, u_t) - c_t(x_t^\pi, u_t^\pi),$$

and we seek to bound this quantity with high probability for a fixed $\pi \in \Pi$. We focus on the benchmark policy class of strongly stable linear policies that choose $u_t = Kx_t$. i.e.,

$$\Pi_{\text{lin}} = \{K \in \mathbb{R}^{d_u \times d_x} \ : \ K \text{ is } (\kappa, \gamma)\text{-strongly stable}\}.$$

We make the following assumptions on our learning problem:

- **Non-stochastic convex and Lipschitz costs.** The costs $c_t$ are arbitrarily determined by an oblivious adversary[4] such that each $c_t(x, u)$ is convex in the pair $(x, u)$ and for any $(x, u), (x', u')$ we have $|c_t(x, u) - c_t(x', u')| \leq \|(x - x', u - u')\|$;[5]

---

[4]An oblivious adversary does not use past random choices of the learner to select its loss functions.
[5]In the full version of the paper [16] we also explain how to accommodate quadratic losses via an appropriate choice of a normalizing constant.

- **i.i.d. Gaussian noise.** $(w_t)_{t=1}^T$ is a sequence of i.i.d. random variables such that $w_t \sim \mathcal{N}(0, \sigma^2 I)$;
- **Stabilizable system.** $A_\star$ is $(\kappa, \gamma)-$strongly stable, and $\|B_\star\| \leq R_B$.

Note that the assumption that $A_\star$ is strongly stable is without loss of generality. Otherwise, given access to a stabilizing controller $K_0$, we show in the full version of the paper [16] a generic black-box reduction that takes any learning algorithm that assumes strongly-stable $A_\star$, augments its observations and adds $K_0 x_t$ to its predicted actions. This essentially replaces $A_\star$ with $A_\star + B_\star K_0$, which is $(\kappa, \gamma)-$strongly stable as desired, and only incurs a $2\kappa$ multiplicative factor in the regret.

The assumption that the noise distribution is known is a standard one that appears in several past works [23, 31, 21, 33]. However some works, e.g., [6, 13, 35], only assume light-tailed (sub-Gaussian or bounded) noise with lower bounded covariance. In the context of this work, the known distribution assumption is used to generate one "fresh" noise sample per time step (see Algorithm 1 and Section 3.3 for further details). We conjecture that recycling past noise estimates could replace the noise generation thus dismissing the known distribution assumption, but leave this as an open problem for future research.

## 2.3 Disturbance Action Policies

We use the, now standard, class of Disturbance Action Policies (DAPs) first proposed by [5]. This class is parameterized by a sequence of matrices $\{M^{[h]} \in \mathbb{R}^{d_u \times d_x}\}_{h=1}^H$. For brevity of notation, these are concatenated into a single matrix $M \in \mathbb{R}^{d_u \times H d_x}$ defined as $M = \left(M^{[1]} \cdots M^{[H]}\right)$. A DAP $\pi_M$ chooses actions

$$u_t = \sum_{h=1}^H M^{[h]} w_{t-h},$$

where recall that the $w_t$ are system disturbances. Consider the benchmark policy class[6]

$$\Pi_{\text{DAP}} = \{\pi_M \; : \; \|M\|_F \leq R_\mathcal{M}\}.$$

There are two main reasons for considering the DAP parameterization. First, the loss functions are convex in $M$, a fact which is generally untrue for $K$ in a linear policy $u_t = K x_t$. This paves the way for tools from the online convex optimization literature. Second, as shown in [5, Lemma 5.2], if $H \in \Omega(\gamma^{-1} \log T)$ and $R_\mathcal{M} \in \Omega(\kappa^2 \sqrt{d_u/\gamma})$ then $\Pi_{\text{DAP}}$ is a good approximation for $\Pi_{\text{lin}}$ in the sense that a regret guarantee with respect to $\Pi_{\text{DAP}}$ gives the same guarantee with respect to $\Pi_{\text{lin}}$ up to a constant additive factor. In light of the above, our regret guarantee will be given with respect to $\Pi_{\text{DAP}}$.

**Bounded memory representation.** As observed in recent literature, the linear dynamics have an infinitely long memory, i.e., all past actions have some effect on the current state, and as such on the losses. However, due to the stability of $A_\star$, the effective memory of the system, $H$, is essentially a constant. To see this, unroll the transition model to get that

$$x_t = A_\star^H x_{t-H} + \sum_{i=1}^H \left(A_\star^{i-1} B_\star u_{t-i} + A_\star^{i-1} w_{t-i}\right) = A_\star^H x_{t-H} + \Psi_\star \tilde{\rho}_{t-1} + w_{t-1}, \tag{1}$$

where $\Psi_\star = [A_\star^{H-1} B_\star, \ldots, A_\star B_\star, B_\star, A_\star^{H-1}, \ldots, A_\star] \in \mathbb{R}^{d_x \times d_\Psi}$ and $\tilde{\rho}_t = [u_{t-H}^\mathsf{T}, \ldots, u_t^\mathsf{T},$ $w_{t-H}^\mathsf{T}, \ldots, w_{t-1}^\mathsf{T}]^\mathsf{T} \in \mathbb{R}^{d_\Psi}$, where $d_\Psi := H d_u + (H-1) d_x$. Now, since $A_\star$ is strongly stable, the term $A_\star^H x_{t-H}$ quickly becomes negligible. Following the notation set by [15], this observation is combined with the DAP policy parameterization to define the following bounded memory representations. Let

$$P(M) = \begin{pmatrix} M^{[H]} & M^{[H-1]} & \cdots & M^{[1]} & & & \\ & M^{[H]} & M^{[H-1]} & \cdots & M^{[1]} & & \\ & & \ddots & \ddots & & \ddots & \\ & & & M^{[H]} & M^{[H-1]} & \cdots & M^{[1]} \\ & & & & I & & \\ & & & & & \ddots & \\ & & & & & & I \end{pmatrix}, \tag{2}$$

---

[6]We note that a more common definition uses $\sum_{h=1}^H \|M^{[h]}\|$ to measure the size of the class. We chose the Frobenius norm for simplicity of the analysis, but replacing it would not change the results significantly.

and for an arbitrary sequence of disturbances $w = \{w_t\}_{t \geq 1}$ define

$$
\begin{aligned}
u_t(M; w) \quad &= \sum_{h=1}^{H} M^{[h]} w_{t-h}; \\
\rho_t(M; w) \quad &= (u_{t+1-H}(M; w)^\mathsf{T}, \ldots u_t(M; w)^\mathsf{T}, w_{t+1-H}, \ldots, w_{t-1})^\mathsf{T} = P(M) w_{t+1-2H:t-1}; \\
x_t(M; \Psi, w) &= \Psi \rho_{t-1}(M; w) + w_{t-1}.
\end{aligned}
\tag{3}
$$

Notice that $u_t, \rho_t, x_t$ do not depend on the entire sequence $w$, but only $w_{t-H:t-1}, w_{t+1-2H:t-1}$, and $w_{t-2H:t-1}$ respectively. Importantly, this means that we can compute these functions with knowledge of only the last (at most) $2H$ disturbances. While this notation does not reveal this fact explicitly, it helps with both brevity and clarity.

---

**Algorithm 1** OCO in Adaptive Linear Control

---

1: **input**: confidence parameter $\delta$, memory length $H$, optimism parameter $\alpha$, regularization parameters $\lambda_\Psi, \lambda_w$, learning rate $\eta_G$, noise bound $W$.
2: **set** $i = 1, \tau = 1, V_1 = \lambda_\Psi I, M_1 = 0$ and $\hat{w}_t = 0, \tilde{w}_t, u_t = 0$ for all $t < 1$ .
3: **define** loss scaling function:

$$
C_M(\Psi) := \sqrt{8} W R_{\mathcal{M}} H \|\Psi\|_F + \alpha \sqrt{2/H} (2 + R_{\mathcal{M}}^{-1} \sqrt{d_x}).
$$

4: **for** $t = 1, 2, \ldots, T$ **do**
5:     **play** $u_t = \sum_{i=1}^{H} M_t^{[h]} \hat{w}_{t-h}$.
6:     **set** $V_{t+1} = V_t + \rho_t \rho_t^\mathsf{T}$ for $\rho_t = (u_{t+1-H}^\mathsf{T}, \ldots, u_t^\mathsf{T}, \hat{w}_{t+1-H}^\mathsf{T}, \ldots, \hat{w}_{t-1}^\mathsf{T})^\mathsf{T}$ .
7:     **observe** $x_{t+1}$ and cost function $c_t$.
8:     **calculate**

$$
(A_t \ B_t) = \underset{(A \ B) \in \mathbb{R}^{d_x \times (d_x + d_u)}}{\arg\min} \sum_{s=1}^{t} \|(A \ B) z_s - x_{s+1}\|^2 + \lambda_w \|(A \ B)\|_F^2, \quad \text{where } z_s = \begin{pmatrix} x_s \\ u_s \end{pmatrix}.
$$

9:     **estimate noise** $\hat{w}_t = \Pi_{\{\|w\| \leq W\}} [x_{t+1} - A_t x_t - B_t u_t]$.
10:     **sample** $\tilde{w}_t \sim \mathcal{N}(0, \sigma^2 I_{d_x})$.
11:     **if** $\det(V_{t+1}) > 2 \det(V_{\tau_i})$ **then**
12:         **start new epoch**: $i = i + 1, \tau_i = t + 1$.
13:         **estimate system parameters**

$$
\Psi_{\tau_i} = \underset{\Psi \in \mathbb{R}^{d_x \times d_\Psi}}{\arg\min} \left\{ \sum_{s=1}^{t} \|\Psi \rho_s - x_{s+1}\|^2 + \lambda_\Psi \|\Psi\|_F^2 \right\}.
$$

14:         **initialize** $\mathcal{A}$ = new instance of $\text{BFPL}_{\delta/6}^\star$ , and set $M_{\tau_i} = \ldots = M_{\tau_i + 2H} = 0$.
15:     **else if** $t \geq \tau_i + 2H$ **then**
16:         **define** expert loss functions:   $\forall k \in [d_\Psi] \times [(2H - 1) d_x], \chi \in \{\pm 1\}$

$$
\tilde{f}_t(M; k, \chi) = c_t(x_t(M; \Psi_{\tau_i}, \tilde{w}), u_t(M; \tilde{w})) - \alpha \sigma \chi \cdot \left( V_{\tau_i}^{-1/2} P(M) \right)_k.
$$

17:         **define** loss vector $\tilde{\ell}_t \in \mathbb{R}^{2(2H-1) d_x d_\Psi^2}$ s.t. $(\tilde{\ell}_t)_{k, \chi} = \tilde{f}_t(M_t(k, \chi); k, \chi) / C_M(\Psi_{\tau_i})$.
18:         **update** experts:   $\forall k \in [d_\Psi] \times [(2H - 1) d_x], \chi \in \{\pm 1\}$

$$
M_{t+1}(k, \chi) = \Pi_{\mathcal{M}} \left[ M_t(k, \chi) - \eta_G \nabla_M \tilde{f}_t(M_t(k, \chi); k, \chi) \right].
$$

19:     **update** prediction $(k_{t+1}, \chi_{t+1}) = \mathcal{A}(\tilde{\ell}_t)$ and **set** $M_{t+1} = M_{t+1}(k_{t+1}, \chi_{t+1})$

---

## 3 Algorithm and Main Result

In this section we present our algorithm for regret minimization in linear systems with unknown dynamics and adversarial convex costs; see Algorithm 1. We denote by $\Pi_{\mathcal{M}}$ the projection onto $\mathcal{M}$.

The algorithm mediates between least squares estimation of the system dynamics (Lines 8 and 13), and optimizing the policy w.r.t. adversarially-changing cost functions. For OCO, the algorithm uses a combination of Online Gradient Descent [38] (Line 18) and $\text{BFPL}_\delta^\star$ [7] (Line 14)—an experts algorithm that also guarantees an overall small number of switches with probability at least $1 - \delta$. Our algorithm uses DAP parameterization (Line 5; see Section 2.3 and notations therein), and feeds the aforementioned online optimization algorithms with lower confidence bounds of the online costs. See below for further details on the algorithm's operation.

We have the following guarantee for Algorithm 1. The proof is deferred to the full version of the paper [16].

**Theorem 2 (Simplified version of Theorem 7 in the full version of the paper [16]).** *Let $\delta \in (0, 1)$ and suppose that we run Algorithm 1 with parameters $R_\mathcal{M}, R_B \geq 1$ and for proper choices of $W, H, \lambda_w, \lambda_\Psi, \eta_G, \alpha$. If $T \geq 8$ then for any $\pi \in \Pi_{\text{DAP}}$, with probability at least $1 - \delta$,*

$$\text{regret}_T(\pi) \leq \text{poly}(\kappa, \gamma^{-1}, \sigma, R_B, R_\mathcal{M}, d_x, d_u, \log(T/\delta))\sqrt{T}.$$

Our algorithm is comprised of multiple components working in tandem. We now give a brief overview of each of the components and how they play together. We note that some components previously appeared in [15], and provide a detailed comparison at the end of the section.

## 3.1 Prerequisites: system estimation and DAP parameterization

*Parameter estimation:* The algorithm proceeds in epochs. At the beginning of each epoch, it estimates the unrolled model via least squares using all past observations (Line 13), and the estimate $\Psi_{\tau_i}$ is then kept fixed throughout the epoch. The epoch ends when the determinant of $V_t$ is doubled (Line 11); intuitively, when the confidence of the unrolled model increases substantially.[7] Throughout the epoch, the algorithm maintains estimates of the transition noise $(\hat{w}_t)_{t=1}^T$ (Lines 8 and 9). We observe that these noise estimates are essentially produced for "free" and no explicit exploration is needed.

*DAP implementation:* While the benefits of $\Pi_{\text{DAP}}$ are clear, notice that it cannot be implemented as is since we do not have access to the system disturbances $w_t$ nor can we accurately recover them (due to the uncertainty in the transition model). Similarly to previous works, our algorithm thus uses estimated disturbances $\hat{w}_t$ to compute its actions. At each time step $t$, the algorithm chooses $u_t$ as a linear function of the past $H$ noise estimates, and parameterized by $M_t$ (Line 5). $M_t$ itself is updated using OCO on surrogate cost functions that are formed as a composition between $c_t(x, u)$ and the bounded memory representations $u_t(M; \hat{w}), x_t(M; \Psi_{\tau_i}, \hat{w})$, implicitly assuming that $M_t$ was kept fixed for the last $H$ time steps. It is therefore crucial that these representations closely reflect the state and action that are actually observed, hence the OCO procedure has to make sure that the sequence $(M_t)_{t=1}^T$ changes slowly (more on this below).

*Construction of lower confidence bounds:* The algorithm uses the estimated unrolled model to minimize regret with respect to lower confidence bounds of the form:

$$c_t(x_t(M; \Psi_{\tau_i}, \hat{w}), u_t(M; \hat{w})) - \alpha' \cdot \|V_{\tau_i}^{-1/2} \rho_{t-1}(M; \hat{w})\|. \tag{4}$$

This lower confidence bound follows immediately by combining the Lipschitzness of $c_t$ and standard self-normalizing concentration bounds [1]. In our analysis, we show that it indeed lower bounds $c_t(x_t, u_t)$. Such lower confidence bounds are used extensively in multi-armed bandit and reinforcement learning literature to efficiently combine exploration and exploitation [10, 11]. Intuitively, their minimization steers the resulting policy towards state-action pairs that either yield low cost, or are insufficiently explored.

## 3.2 Key idea: making the algorithm efficient

The functions in Eq. (4) are, unfortunately, nonconvex (being a difference of two convex functions), and thus cannot be used in OCO algorithms in their current form. However, we overcome this by relaxing the functions in Eq. (4); we do so in two steps. First, we move to an expected, amortized notion of optimism. We can do this since since $\hat{w} \approx w$, which are i.i.d, and thus standard concentration arguments imply that the realized bonus term is close to its conditional expectation, which takes the form:

$$\sqrt{\mathbb{E}\|V_{\tau_i}^{-1/2} \rho_{t-1}(M; w)\|^2} = \sigma\|V_{\tau_i}^{-1/2} P(M)\|_F,$$

---

[7]Concretely, the volume of the confidence ellipsoid around the unrolled model decreases by a constant factor.

where $P(M)$ is define in Eq. (2). Second, building on a trick from [22] in the context of linear bandit optimization, we further bound $\left\|V_{\tau_i}^{-1/2} P(M)\right\|_F \le d_\Psi \left\|V_{\tau_i}^{-1/2} P(M)\right\|_\infty$ (where $\|\cdot\|_\infty$ is the entry-wise matrix infinity norm). Due to an adaptivity issue (more on this below), we also replace the estimated noises $\hat{w}$ in the cost term with random simulated noises $\tilde{w} \sim \mathcal{N}(0, \sigma^2 I)$. After this relaxation, the resulting optimistic cost can be written as a minimum of convex functions of the form

$$\tilde{f}_t(M; k, \chi) = c_t(x_t(M; \Psi_{\tau_i}, \hat{w}), u_t(M; \hat{w})) - \alpha \sigma \chi \cdot \left(V_{\tau_i}^{-1/2} P(M)\right)_k, \tag{5}$$

where $k \in [d_\Psi] \times [(2H-1)d_x]$ is a matrix coordinate , $\chi \in \{\pm 1\}$ is a sign variable , and $\alpha = d_\Psi \alpha'$. Crucial to this trick is the fact that, unlike Eq. (4), the linearized non-convex term is independent of the time index $t$. This observation yields computationally-efficient regret minimization via a two-tier approach described as follows. We run a different copy of Online Gradient Descent [38] for each value of $k, \chi$, maintaining a different set of DAP parameters $M_t(k, \chi)$, and fed with $\tilde{f}_t(\cdot; k, \chi)$ (Line 18). On top of the OGD algorithms, we run an experts meta-algorithm to minimize $\tilde{f}_t(M_t(k, \chi); k, \chi)$ over $k, \chi$ (Line 19), treating the output of each OGD algorithm as an expert.

Observe that having initially taken expectation over the noises yields an exploration bonus term that, for fixed $M$, is fixed throughout each epoch. This makes sure that our OCO algorithms, that are restarted at every epoch, can compare against $M_\star$ (the best in hindsight) with $k$ and $\chi$ being fixed at the start of the epoch.

### 3.3  Additional challenges

*Stabilizing the meta-algorithm:* Our hedging approach nevertheless comes at a price. The choices of the meta-algorithm are inherently random, thus $M_t$ might change abruptly between consecutive rounds (recall that DAP require slowly-changing $M_t$). We therefore use a version of Follow the Lazy Leader (BFPL$^\star$; [7]) that guarantees, with high probability, both no-regret and a small number of switches. The small number of switches in conjunction with the fact that each of the expert algorithms generate slowly-changing decisions, guarantee that $M_t$ itself is slowly-changing overall.

*Mitigating adaptivity in costs:* Even so, the guarantees of BFPL$^\star$ hold only against *oblivious* adversaries (and this limitation is inherent, as [7] discuss extensively), yet the loss sequence constituting of the functions in Eq. (5) is unfortunately *not* oblivious. This is because the noise estimate $\hat{w}$ were generated using policies derived from previous choices of BFPL$^\star$. We overcome this hindrance relying on the fact that the noise vectors are drawn from a known (Gaussian) distribution. This allows to sample i.i.d. copies of the noise vectors $\tilde{w}$ (Line 10) that we use in $\tilde{w}$ instead of $\hat{w}$, arriving at the functions defined in Line 16, and ensuring that BFPL$^\star$ receives obliviously-generated losses.

**Comparison with [15].** Algorithm 1 here shares some components with Algorithm 1 of [15]. In particular, the parameter estimation and DAP implementation are identical. However, there are some crucial differences. Most notably, the optimistic cost minimization must handle the non-stationary adversarial costs rather than the stationary stochastic ones. As such Algorithm 1 employs a regret minimization scheme with respect to the non-convex optimistic costs (Section 3.2) as opposed to a simpler empirical risk minimization (ERM) in [15]. The stability and adaptivity challenges above are also unique to our adversarial setting. In [15] these are avoided by changing the decision only a small (logarithmic) number of times, which can only be done for stochastic costs.

## 4  Analysis

In this section we give a (nearly) complete proof of Theorem 2 in a simplified setup, inspired by [33], where $A_\star = 0$. The analysis in the general case is significantly more technical and thus deferred from this extended abstract (see the full version of the paper [16] for full details).

Suppose that $A_\star = 0$ and thus $x_{t+1} = B_\star u_t + w_t$, assume that $c_t(x, u) = c_t(x)$, i.e., the costs do not depend on $u$, and aim to minimize the pseudo regret,

$$\max_{u: \|u\| \le R_u} \sum_{t=1}^{T} [J_t(B_\star u_t) - J_t(B_\star u)],$$

where $J_t(x) = \mathbb{E}_w c_t(x + w)$, is the expected instantaneous cost, which can be computed from $c_t(x)$ for a known noise distribution. The resulting problem is an instance of the following variant of online convex optimization, which we now define with clean notation as to avoid confusion with our general setting.

## 4.1 Simplified setting: OCO with a Hidden Linear Transform

Consider the following setting of online convex optimization. Let $\mathcal{S} \subseteq \mathbb{R}^{d_a}$ be a convex decision set. (We denote by $\Pi_{\mathcal{S}}$ the projection onto $\mathcal{S}$.) At round $t$ the learner:

(i) predicts $a_t \in \mathcal{S}$;

(ii) observes cost function $\ell_t : \mathbb{R}^{d_y} \to \mathbb{R}$ and state $y_{t+1} = Q_\star a_t + \epsilon_t$;

(iii) incurs cost $\ell_t(Q_\star a_t)$.

We have that $\epsilon_t \in \mathbb{R}^{d_y}$ are i.i.d. noise terms, $Q_\star \in \mathbb{R}^{d_y \times d_a}$ is an unknown linear transform, and $y_t \in \mathbb{R}^{d_y}$ are noisy observations. The cost functions are chosen by an oblivious adversary, and we consider minimizing the regret, defined as

$$\mathrm{regret}_T = \max_{a \in \mathcal{S}} \sum_{t=1}^{T} [\ell_t(Q_\star a_t) - \ell_t(Q_\star a)].$$

**Assumptions.** We make the following assumptions:

- $\ell_t(\cdot)$ are convex and $1$–Lipschitz;

- There exist known $W, R_Q \geq 0$ such that $\|\epsilon_t\| \leq W$, and $\|Q_\star\| \leq R_Q$.

- For all $a \in \mathcal{S}$ we have $\|a\| \leq R_a/2$.

---

**Algorithm 2** OCO with a hidden linear transform

1: **input:** optimism parameter $\alpha$, regularizer $\lambda$, learning rates $\eta_G, \eta_M$
2: **set:** $V_1 = \lambda I, \widehat{Q}_1 = 0, i = 1, \tau_1 = 1$, and $a_1(k, \chi) \in \mathcal{S}, p_t(k, \chi) = 1/2d_a \ \forall k \in [d_a], \chi \in \{\pm 1\}$.
3: **for** $t = 1, 2, \ldots, T$ **do**
4:     **draw** $(k_t, \chi_t) \sim p_t$, and **play** $a_t = a_t(k_t, \chi_t)$.
5:     **observe** $y_{t+1} = Q_\star a_t + w_t$ and cost function $\ell_t$, and **set** $V_{t+1} = V_t + a_t a_t^\mathsf{T}$.
6:     **if** $\det(V_{t+1}) > 2\det(V_{\tau_i})$ **then**
7:         **start new episode** $i = i+1, \tau_i = t+1$, and set $p_{t+1}(k, \chi) = 1/2d_a, a_{t+1}(k, \chi) = a_t(k, \chi)$.
8:         **estimate** parameters: $\widehat{Q}_{\tau_i} = \arg\min_{Q \in \mathbb{R}^{d_y \times d_a}} \sum_{s=1}^{t} \{\|Q a_s - y_{s+1}\|^2 + \lambda \|Q\|_F^2\}$.
9:     **else**
10:         **define** expert loss functions: $\bar{\ell}_t(a; k, \chi) = \ell_t(\widehat{Q}_{\tau_i} a) - \alpha\chi \cdot (V_{\tau_i}^{-1/2} a)_k$.
11:         **update** experts: $a_{t+1}(k, \chi) = \Pi_{\mathcal{S}}[a_t(k, \chi) - \eta_G \nabla_a \bar{\ell}_t(a_t(k, \chi); k, \chi)]$.     ▷ OGD
12:         **update** prediction: $p_{t+1}(k, \chi) \propto p_t(k, \chi) \exp(-\eta_M \bar{\ell}_t(a_t(k, \chi); k, \chi))$.     ▷ MW

---

**Algorithm.** Our algorithm for this simplified setup is detailed in Algorithm 2. Unlike the full control setting, the adversarial costs here have no memory, thus enable the following simplifications compared to Algorithm 1. First, we can forgo the DAP parameterization and directly optimize the prediction $a_t$. This both removes the need to estimate the disturbances, and simplifies the construction of the lower confidence bound. Moreover, the lack of memory obviates the need to make our predictions change slowly over time, and we replace the BFPL$^\star$ sub-routine with Multiplicative Weights (MW) [see 8].

## 4.2 Analysis

The main result of this section bounds the regret of Algorithm 2 with high probability.

**Theorem 3.** *Let $\delta \in (0, 1)$ and suppose that we run Algorithm 2 with parameters*

$$\eta_G = \frac{R_a}{(2\alpha R_a^{-1} + R_Q)\sqrt{T}}, \eta_M = \frac{\sqrt{\log(2d_a)}}{2(2\alpha + R_a R_Q)\sqrt{T}}, \lambda = R_a^2, \alpha = \sqrt{d_a}\left(W d_y \sqrt{8\log\frac{2T}{\delta}} + \sqrt{2}R_a R_Q\right).$$

*If $T \geq 8$ then with probability at least $1 - \delta$,*

$$\mathrm{regret}_T \leq 77 d_a^{3/2}\left(W d_y \sqrt{8\log\frac{2T}{\delta}} + R_a R_Q\right)\sqrt{T \log^2 \frac{4 d_a T^2}{\delta}}.$$

The proof of Theorem 3 is composed of two main lemmas. Similarly to the control setting, we first define an optimistic loss

$$\bar{\ell}_t(a) = \ell_t(\widehat{Q}_{\tau_{i(t)}} a) - \alpha \|V_{\tau_{i(t)}}^{-1/2} a\|_\infty,$$

where $i(t) = \max\{i : \tau_i \le t\}$. The following lemma shows that the optimistic loss lower bounds the true loss, and bounds the error between the two.

**Lemma 4 (optimism).** *Suppose that* $\sqrt{d_a}\|\widehat{Q}_{\tau_{i(t)}} - Q_\star\|_{V_{\tau_{i(t)}}} \le \alpha$. *Then for any* $a \in \mathbb{R}^{d_a}$,

$$\bar{\ell}_t(a) \le \ell_t(Q_\star a) \le \bar{\ell}_t(a) + 2\alpha \sqrt{a^\mathsf{T} V_{\tau_{i(t)}}^{-1} a}.$$

**Proof.** The proof follows standard arguments (see e.g. Lemma 3 in [15]). We first use the Lipschitz assumption to get

$$\begin{aligned}
|\ell_t(Q_\star a) - \ell_t(\widehat{Q}_{\tau_{i(t)}} a)| &\le \|(Q_\star - \widehat{Q}_{\tau_{i(t)}})a\| \\
&\le \|Q_\star - \widehat{Q}_{\tau_{i(t)}}\|_{V_{\tau_{i(t)}}} \|V_{\tau_{i(t)}}^{-1/2} a\| \\
&\le \frac{\alpha}{\sqrt{d_a}}\|V_{\tau_{i(t)}}^{-1/2} a\| \\
&\le \alpha\|V_{\tau_{i(t)}}^{-1/2} a\|_\infty,
\end{aligned}$$

where the second and third transitions also used the estimation error and that $\|a\| \le \sqrt{d_a}\|a\|_\infty$. We thus have on one hand,

$$\ell_t(Q_\star a) \ge \ell_t(\widehat{Q}_{\tau_{i(t)}} a) - \alpha\|V_{\tau_{i(t)}}^{-1/2} a\|_\infty = \bar{\ell}_t(a),$$

and on the other hand we also have

$$\ell_t(Q_\star a) \le \ell_t(\widehat{Q}_{\tau_{i(t)}} a) + \alpha\|V_{\tau_{i(t)}}^{-1/2} a\|_\infty = \bar{\ell}_t(a) + 2\alpha\|V_{\tau_{i(t)}}^{-1/2} a\|_\infty \le \bar{\ell}_t(a) + 2\alpha\sqrt{a^\mathsf{T} V_{\tau_{i(t)}}^{-1} a},$$

where the last step also used $\|a\|_\infty \le \|a\|$. ∎

Next, the following result bounds the regret with respect to the optimistic cost functions.

**Lemma 5.** *Define* $G_i = \|\widehat{Q}_{\tau_i}\| + \alpha\lambda^{-1/2}$ *and* $\bar{G} = 2\alpha\lambda^{-1/2} + R_Q$. *With probability at least* $1 - \delta$, *for all epochs* $i \ge 1$ *simultaneously:*

$$\sum_{t=\tau_i}^{\tau_{i+1}-1} \left(\bar{\ell}_t(a_t) - \bar{\ell}_t(a)\right) \le 3R_a\left(\bar{G} + \bar{G}^{-1}G_i^2\right)\sqrt{T\log\frac{2d_a T^2}{\delta}}.$$

**Proof.** First, fix an epoch $i$ and notice that $a_t(k, \chi)$ are the result of running Online Gradient Descent (OGD) on the functions $\bar{\ell}_t(\cdot; k, \chi)$, which are $G_i$ Lipschitz. A classic regret bound for OGD (see Lemma 25 in the full version of the paper [16]) then gives us that for all $a \in \mathcal{S}$ and $\tau_i \le s \le T$

$$\sum_{t=\tau_i}^{s} \bar{\ell}_t(a_t(k, \chi); k, \chi) - \bar{\ell}_t(a; k, \chi) \le \frac{1}{2}R_a(\bar{G} + G_i^2\bar{G}^{-1})\sqrt{T}.$$

Next, note that MW is invariant to a constant shift in the loss vectors. Letting $a_0 \in \mathcal{S}$ be arbitrary, we have that $p_t$ is updated according to the MW rule with the loss of each expert being $\bar{\ell}_t(a_t(k, \chi); k, \chi) - \ell_t(\widehat{Q}_{\tau_i} a_0)$. Using the Lipschitz property of $\ell_t$, these are bounded as

$$|\bar{\ell}_t(a_t(k, \chi); k, \chi) - \ell_t(\widehat{Q}_{\tau_i} a_0)| \le \|\widehat{Q}_{\tau_i}\|\|a_t(k, \chi) - a_0\| + \alpha\lambda^{-1/2}\|a_t(k, \chi)\| \le G_i R_a.$$

A standard regret guarantee of MW (Lemma 26 in the full version of the paper [16]) thus gives us that with probability at least $1 - \delta$,

$$\sum_{t=\tau_i}^{s} \bar{\ell}_t(a_t(k_t, \chi_t); k_t, \chi_t) - \bar{\ell}_t(a_t(k, \chi); k, \chi) \le R_a\left(\bar{G} + \bar{G}^{-1}G_i^2\right)\sqrt{6T\log\frac{2d_a T}{\delta}},$$

for all $k \in [d_a]$, $\chi \in \{-1, 1\}$, and $\tau_i \le s \le T$.

Now, let $k_t^*(a), \chi_t^*(a)$ be such that $\bar{\ell}_t(a) = \bar{\ell}_t(a; k_t^*(a), \chi_t^*(a))$ for all $\tau_i \le t < \tau_{i+1}$. Importantly, notice that $k_t^*(a), \chi_t^*(a)$ are independent of the time index $t$. This is because the minimum in $\bar{\ell}_t$ is taken over the optimism term, which is independent of $t$ inside a given epoch. For ease of notation, the following will omit the dependence of $k^*, \chi^*$ on $a$, which will be kept as a fixed (arbitrary) comparator. Combining the above, with probability $\ge 1 - \delta$ we have that for all $a \in \mathcal{S}$:

$$\sum_{t=\tau_i}^{\tau_{i+1}-1} \bar{\ell}_t(a_t) - \bar{\ell}_t(a) \le \sum_{t=\tau_i}^{\tau_{i+1}-1} \bar{\ell}_t(a_t; k_t, \chi_t) - \bar{\ell}_t(a; k^*, \chi^*) \qquad (\bar{\ell}_t(\cdot) \le \bar{\ell}_t(\cdot; k, \chi))$$

$$= \sum_{t=\tau_i}^{\tau_{i+1}-1} \big( \bar{\ell}_t(a_t(k_t, \chi_t); k_t, \chi_t) - \bar{\ell}_t(a_t(k^*, \chi^*); k^*, \chi^*) \big)$$

$$+ \sum_{t=\tau_i}^{\tau_{i+1}-1} \big( \bar{\ell}_t(a_t(k^*, \chi^*); k^*, \chi^*) - \bar{\ell}_t(a; k^*, \chi^*) \big)$$

$$\le 3R_a \big( \bar{G} + \bar{G}^{-1} G_i^2 \big) \sqrt{T \log \frac{2 d_a T}{\delta}}.$$

Repeating the above with $\delta/T$ and taking a union bound over the epochs (of which there are at most $T$) concludes the proof. ∎

We are now ready to prove Theorem 3. We focus here on the main ideas, deferring some details to the full version of the paper [16].

**Proof of Theorem 3.** We decompose the regret as

$$\mathrm{regret}_T(a) \le \underbrace{\sum_{t=1}^{T} \ell_t(Q_\star a_t) - \bar{\ell}_t(a_t)}_{R_1} + \underbrace{\sum_{t=1}^{T} \bar{\ell}_t(a_t) - \bar{\ell}_t(a)}_{R_2} + \underbrace{\sum_{t=1}^{T} \bar{\ell}_t(a) - \ell_t(Q_\star a)}_{R_3},$$

and conclude the proof by bounding each term on the following good event. Suppose Lemma 5 holds for all epochs with $\delta/2T$, and that $\sqrt{d_a} \|\widehat{Q}_t - Q_\star\|_{V_t} \le \alpha$ for all $t \le T$, which follows from a standard least squares estimation bound (see Lemma 22). Taking a union bound, this event holds with probability at least $1 - \delta$. We conclude that Lemma 4 holds and thus $R_3 \le 0$. Moreover, we get that

$$R_1 \le \sum_{i=1}^{N} \sum_{t=\tau_i}^{\tau_{i+1}-1} 2\alpha \sqrt{a_t^\mathsf{T} V_{\tau_i}^{-1} a_t} \le 2\alpha \sum_{t=1}^{T} \sqrt{2 a_t^\mathsf{T} V_t^{-1} a_t} \le 2\alpha \sqrt{2T \sum_{t=1}^{T} a_t^\mathsf{T} V_t^{-1} a_t} \le 2\alpha \sqrt{10 T d_a \log T},$$

where the second inequality uses Lemma 27 of [21], which states that for $V_1 \succeq V_2 \succeq 0$ we have $V_1 \preceq V_2(\det(V_1)/\det(V_2))$, the third is due to Jensen's inequality, and the fourth is a standard algebraic argument (see Lemma 23).

Now, an immediate corollary (see Eq. (10)) of the least square error bound is that $\|\widehat{Q}_t\| \le \alpha \lambda^{-1/2} + R_Q$. We thus have that $G_i \le \bar{G}$ for all $i \le N$. Next, notice that the number of epochs satisfies $N \le 2d_a \log T$ (Lemma 24). We conclude that

$$R_2 = \sum_{i=1}^{N} \sum_{t=\tau_i}^{\tau_{i+1}-1} \big( \bar{\ell}_t(a_t) - \bar{\ell}_t(a) \big) \le \sum_{i=1}^{N} 3R_a \big( \bar{G} + \bar{G}^{-1} G_i^2 \big) \sqrt{T \log \frac{4 d_a T^2}{\delta}} \qquad \text{(Lemma 5)}$$

$$\le 12 d_a \big( 2\alpha + R_a R_Q \big) \sqrt{T \log^2 \frac{4 d_a T^2}{\delta}}. \qquad \blacksquare$$

## Acknowledgments and Disclosure of Funding

This work was partially supported by the Israeli Science Foundation (ISF) grant 2549/19, by the Len Blavatnik and the Blavatnik Family foundation, by the Yandex Initiative in Machine Learning, and by the Israeli VATAT data science scholarship. AC is supported by the Israeli Science Foundation (ISF) grant no. 2250/22.

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
