# OpenReview forum: "Rate-Optimal Online Convex Optimization in Adaptive Linear Control"
_NeurIPS.cc/2022/Conference — NeurIPS 2022 Accept_

### Official Review · Reviewer_U5w9 · 2022-07-07

**Rating:** 7
**Confidence:** 3
**Soundness:** 3 good
**Presentation:** 3 good
**Contribution:** 3 good

**Summary:**

The authors study the problem of online control of an unknown linear dynamical system w.r.t. a sequence of (oblivious) adversarial cost functions. They give a computationally efficient algorithm that achieves $O(\sqrt{T})$ regret, compared to the best static stabilizing controller in hindsight. Their approach does not require strong convexity of the cost functions as in prior work.

**Questions:**

- As pointed out by the authors, one drawback of the DAP policy class is that it requires the noise disturbance sequence $\{w_t\}$ to compute the action $u_t$. An alternative parameterization is the system level synthesis framework (see Section 2 of Anderson et al. https://www.sciencedirect.com/science/article/abs/pii/S1367578819300215), which is essentially the DAP parameterization with an extra subspace constraint involving the dynamics. By adding the subspace constraint (which can be encoded using the current $(A_t, B_t)$), one can then turn the DAP policy into a proper state feedback policy. I wonder if this parameterization would simplify various aspects of both the algorithm (no need to estimate the noise sequences in Line 8), and the proof.

**Limitations:**

- As mentioned previously, no experiments. In my opinion, condensing Section 4 and adding an experimental evaluation would further strengthen this paper from accept to strong accept.

**Strengths And Weaknesses:**

**Strengths:**
- Paper is well written.
- Authors advance the state of the art in online LQR control.
- Proof techniques may be of independent interest.

**Weaknesses:**
- The algorithm is not the most straightforward, and involves a lot of different components (each of the components are, however, standard in the OCO literature).
- No experiments showing that the algorithm is actually practical.
- I would have liked to see a much more detailed comparison to Cassel et al. [16]. The setting of [16] is stochastic (instead of adversarial costs), but Algorithm 1 from [16] seems similar. The main difference is that the OFU step in [16] is replaced with the BFPL step in Lines 15-18 of Algorithm 1. Is this for computational reasons only, or is this actually necessary? (Apologies if this is an obvious question, I am not an expert in online convex optimization). This is also where an experiment would have been nice, to see the practical difference between these two algorithms.

---

> ### Author Response · Authors · 2022-08-01
> **response**
>
> * “No experiments…”: While experiments could certainly add interest, the main focus of our work is theoretical. Additionally, we are not aware of any established benchmarks in recent literature. Establishing such benchmarks would be an important undertaking.
> * Comparison with Cassel et al. [16]: The difference between the algorithms is not only due to computational considerations, but rather due to a crucially different problem setting. In our setup, the costs are adversarial and ensuring low regret requires a different algorithmic approach (OCO) that is more robust to non-stationary, long term variations in the costs. This yields two main challenges: making the surrogate cost construction oblivious (which is done using noise sampling), and creating an OCO method tailored to the particular non-convex structure of the problem. Removing the computational considerations would have only slightly simplified the description of the algorithm. This is mostly explained in sections 3.1 and 3.2. In light of your comment we will revise the discussion to more directly compare with [16].
> * System Level Synthesis parameterization: while this is an interesting approach, we suspect it will not simplify our results. In both DAP and SLS we need to estimate A,B. In the former these are used to estimate the noise and in the latter - the subspace constraint. Since the noise is stochastic, its estimation is “free” in the sense that we do not need to actively explore the state and action spaces to get good estimates. In contrast, the additional cost due to system estimation in SLS (at least as presented in Anderson et al. Theorem 2.2 and discussion thereafter) depends on $\||\hat{A}-A\||, \||\hat{B}-B\||$, which require uniform exploration to keep small (e.g., adding Gaussian perturbation to the actions). Unfortunately, we are not aware of any way to obtain $\sqrt{T}$ regret using uniform exploration and indeed most of the technical challenge in our work is devoted to the implementation of efficient non-uniform exploration.

---

> > ### Comment · Reviewer_U5w9 · 2022-08-06
> > **thanks for the response**
> >
> > Thanks for clarifying the comparison with [16], and highlighting why an SLS based approach would not be a significant win.
> >
> > My recommendation stands: While I still think experiments would only make the paper more compelling (addressing the practicality of the proposed algorithm), the theoretical contributions are significant enough to merit acceptance.

---

### Official Review · Reviewer_F712 · 2022-07-11

**Rating:** 7
**Confidence:** 3
**Soundness:** 3 good
**Presentation:** 3 good
**Contribution:** 3 good

**Summary:**

The paper looks at the problem of controlling a linear dynamical system when there are adversarial convex cost functions. They study this problem in the setting when there is full feedback on the cost and the current state of the system. They provide an algorithm that obtains optimal regret even when the cost functions are not strongly convex. They also provide a non-convex lower bound for the cost.

**Questions:**

What are the challenges of moving away from a full feedback model to a limited feedback model? What if you were to observe the state and/or the cost function, with some form of error? How would this affect the tractability of the problem?

**Limitations:**

The paper points out that the meta-algorithms they use as experts have inherent randomness that could result in instability. They point out that the use of a follow the lazy leader algorithm helps stabilize the effects. They state that there are inherent challenges of using the follow the lazy leader algorithm as it does not work well against non-oblivious adversaries. They do get over this barrier by using gaussian distribution assumptions on noise.

**Strengths And Weaknesses:**

- Originality: The problem of controlling linear dynamical systems has been studied in the literature. The paper attempts to further this study by considering the model of adversarial and general convex cost functions. They provide a computationally efficient algorithm that gives near-optimal regret.

- Quality and Clarity: The paper is well written and clear.

- Significance:  The paper claims to provide the first computationally efficient algorithm that achieves near-optimal regret for the linear dynamical system when the cost functions change over time.

---

> ### Author Response · Authors · 2022-08-01
> **response**
>
> Partial feedback: This is indeed a very interesting question, yet a very difficult one as even the exact full info case already required overcoming significant challenges. Concretely, if we assume bandit feedback of the costs, we obtain a setup that is far more general than Bandit Convex Optimization (BCO), which is an immensely challenging setting in its own right. On a slightly more positive note, we suspect that our algorithmic components are robust to mild perturbations in their inputs. If so, while still challenging, it might be possible to accommodate weaker forms of partial feedback using our approach.

---

> > ### Comment · Reviewer_F712 · 2022-08-06
> > **Thanks for the response**
> >
> > Thank you for your response. I will retain my score of 7 as I feel this is a sufficiently good paper for an acceptance.

---

### Official Review · Reviewer_jGE4 · 2022-07-14

**Rating:** 7
**Confidence:** 4
**Soundness:** 4 excellent
**Presentation:** 3 good
**Contribution:** 3 good

**Summary:**

The paper provides a high-probability $\tilde{O}(\sqrt{T})$ regret bound for online linear control with convex, Lipshitz costs and stochastic noise with known covariance provided a stabilizing controller.  This is in contrast to existing work with $O(T^{2/3})$ regret for convex costs or $\tilde{O}(\sqrt{T})$ with restrictions on costs like strong convexity.

**Questions:**

Could notation be described earlier before the algorithm?

**Limitations:**

The authors could do a better job of relating the assumptions of this work to other related works.

**Strengths And Weaknesses:**

Strengths:
- Online linear control is a classically important problem and policy regret approaches to the problem have been popular in the last few years.  Improved rates are a worthwhile problem.
- The approach is technically challenging, combining lower confidence bound techniques, online control policies, along with slowly changing experts.  These cannot be naively combined as dealing with the nonconvexity of the lower confidence bounds requires cleverness.
- The exposition of the algorithm/proof are relatively easy to understand and follow

Weaknesses:
- Much of the recent work in online control focused on regret-minimization allows for adversarial bounded costs.  Stochastic noise with known covariance is much weaker.
- Stabilizing controller is required for this method while some other methods are fully black box.
- Some parts of the writing can be a bit hard to follow notation wise.  For example, understanding that $k$ was a tuple as defined in the equation 3 required reference to the algorithm.

---

> ### Author Response · Authors · 2022-08-01
> **response**
>
> * Adversarial vs. Stochastic noise: Adversarial noise is an interesting yet significantly more challenging setting. Indeed, it is not clear how to adaptively explore the system parameters in this case, and previous works resolved to a simple explore then exploit strategy that, for general convex costs, yields a $T^{2/3}$ regret rate. In the stochastic noise setting, we point out that our assumption (stochastic noise from a known distribution) already appeared in several past works (Dean et al. ‘18, Cohen et al.’19, Agarwal et al. ‘19, Plevrakis and Hazan, ‘20, and others). We suspect that removing the assumption will be highly technically challenging, and indeed an interesting direction for future research.
> * Stabilizing controller: We note that assuming access to an initial stabilizing controller is standard in most recent literature. When such a controller is not available, it can typically be recovered via a fixed warmup period (see e.g. Chen and Hazan, "Black-box control for linear dynamical systems"), but the cost is exponential in the dimension (yet independent of T) and cannot generally be avoided (see lower bound in the same work).
> * “...a bit hard to follow notation wise”: Thanks for pointing this out. We will do a better job describing our notations in the final version of our paper. In particular, we will define $k$ outside of the algorithm.
> * “...relating the assumptions of this work to other related works”: In light of your comment, we will further clarify how our assumptions compare with previous works.

---

> > ### Comment · Reviewer_jGE4 · 2022-08-03
> > **Updated Score**
> >
> > Thanks for addressing my concerns.  I have increased my score (was previously on 6/7 boundary) as I think this is a strong paper and should definitely be accepted with some minor clarity improvements.

---

> > > ### Author Response · Authors · 2022-08-04
> > > **Thanks for the response**
> > >
> > > Thank you for your response and the score update.  We are glad to have been able to address your concerns - we will definitely work on improving our presentation using your remarks and the comments from the other reviews.

---

### Official Review · Reviewer_2qLS · 2022-07-25

**Rating:** 7
**Confidence:** 4
**Soundness:** 4 excellent
**Presentation:** 4 excellent
**Contribution:** 3 good

**Summary:**

This paper considers the task of controlling an unknown (stochastic) linear dynamical system subject to changing convex costs revealed online. The main result is a computationally efficient achieving T^0.5 regret against the class of stable linear policies.

**Questions:**

I understand the need to play slow-moving iterates. But neither once-randomized FTPL nor shrinking dartboard is necessary here, to the best of my comprehension, because the underlying decision space is continuous, obviating the need for explicit sampling. Here is a simpler alternative.

Given M_1, … M_K expert recommendations, and (normalized) weights from a MW instance p_1, .. p_K, play the policy M’=sum_i p_i M_i. By convexity of the true cost, cost(M’) < sum_i p_i cost(M_i). Furthermore, p_i’s change slowly as usual. I point out MW for simplicity, but any low-regret algorithm with slow moving iterates is equally suitable (e.g. OGD or the expected iterate for FTPL).

In fact, this approach has potentially greater technical merit in that it also works for adaptive adversaries.

Can the authors comment on this consideration?

**Limitations:**

Yes

**Strengths And Weaknesses:**

— The previously best known regret bound was T^2/3, improvable to T^0.5 for strongly convex costs or in the stochastic setting. This is an unambiguous improvement.

— The primary innovation here is choosing the compatible efficient implementation of OFU to combine with OGD/regret-minimizing algorithm. The typically non-convex optimistic cost can be convexified with the correct guess of the active norm constraint; this was used in the recent result (CCK’22) on stochastic costs.

— Here the natural choice would be to use a regret minimizer to guess this active constraint, except such a procedure would only be sound if the cost associated the norm penalty was stationary, i.e. when the best in hindsight decision coincides with the point-wise best. This stationarity is what the paper notes (Line 211). Notably, this characterization was also true, but not necessary for the result, in the stochastic case (CCK’22). I think this observation is neat.

— The paper is well written, and, to a great extent, linearly readable.

Setting apart the FTPL/switching issue (see below), I do not have any reservations on the claims made.

---

> ### Author Response · Authors · 2022-08-01
> **response**
>
> Thank you for your keen interest in our work. Your suggestion of playing the convex combination of the experts as opposed to sampling an expert is very sensible and, indeed, one we have previously considered. Notice that the online algorithm operates on the optimistic costs, which are non-convex, and thus Jensen’s inequality does not hold. This means that although the convex combination of predictions is slowly moving, its cost may be prohibitively large. We have attempted other methods to make this concept work, for example, using Jensen’s on the original costs before moving to their optimistic versions, however, these also failed due to similar reasons. This has led us to abandon this direction, but we would be happy to hear if you have further insights!

---

> > ### Comment · Reviewer_2qLS · 2022-08-09
> > **Update**
> >
> > Hi, thanks for the response.
> >
> > I'm not fully convinced that using convexity + Jensen on the true cost does not work; I understand that the optimistic cost is a non-starter. However, this is something I need to spend some time on to convince myself or disabuse myself of this idea. My score (or possible increments) are in any case not reliant on this prompt succeeding or failing to materialize.
> >
> > As stated in my original review, I think the work presents an unambiguous improvement over what we known. I am happy to vote to accept, and make a case for this paper if the need arises.

---

### Meta-Review · Area_Chair_RbGc · 2022-08-21

**Recommendation:** Accept
**Confidence:** Certain

**Metareview:**

I recommend acceptance due to the unabiguously and uniformly positive opinions of the reviewers. The reviews identify the authors' clear contribution (improving the rate) for a problem of interest in the community (online linear control with relaxed assumptions). Concerns about clarity or alternative approaches were resolved during the discussion period.

**Award:**

No

---

### Decision · Program_Chairs · 2022-09-14

Accept